

# Impact of forest fragmentation on river water quality: an example from a typical subtropical hilly basin

Biao Li[1,2], Xiaolei Huang[3], Qiang Zhong[3] and Xiuxiu Wu[3]

[1] National Engineering Laboratory for Applied Technology of Forestry & Ecology in Southern China, Central South University of Forestry and Technology, Changsha, Hunan, China
[2] College of Life and Environmental Sciences, Central South University of Forestry and Technology, Changsha, Hunan, China
[3] School of Public Health and Health Management, Gannan Medical University, Ganzhou, Jiangxi, China

Corresponding author
Xiuxiu Wu, wuxiuxiu@gmu.edu.cn

## ABSTRACT

**Background:** Forest fragmentation, driven by natural and human activities, is increasing. However, the impact of forest fragmentation on river water quality remains ambiguous.

**Methods:** In this study, water quality data were collected from 15 monitoring sites in the upper Ganjiang River basin in winter and summer, and the forest landscape fragmentation metrics in the sub-basin was calculated to assess its seasonal impact on river water quality.

**Results:** The results indicated that water quality in the area is generally satisfactory, with total nitrogen (TN) as the main pollutant. Redundancy analysis (RDA) showed that the explanation rate of the six forest landscape fragmentation metrics to the water quality change in summer was 41.21%, and in winter, their explanation rate of water quality change increased by 14.26%. Among them, the effective mesh size (MESH) was negatively correlated with most river water quality indicators, with a contribution rate of 20.9%. While the interspersion and juxtaposition index (IJI) was positively correlated with most water quality indicators in winter, with a contribution rate of 44.9%. It is worth noting that the thresholds for IJI and MESH of forest were the same in winter and summer, 28.1% and 7.89e+0.5ha, respectively, when the probability of an abrupt change in TN concentration reached 100%. This is implied that when the adjacency of forest patches is less than 28.1% and the connectivity of forest patches is more than 7.89e+0.5ha, it may contribute to the reduction of TN concentration in rivers. These findings provide valuable insights into how varying degrees of forest fragmentation can lead to deterioration in river water quality, and allow for further planning of forest structure based on forest fragmentation thresholds to improve regional water quality.

## INTRODUCTION

Forests have a prominent role to perform in the global water cycle (*Sun et al., 2024*; *Vose, 2019*). On the one hand, forests have a strong capacity to conserve water because of their

multi-layered structure (tree layer, shrub layer, herb layer, moss lichen layer, and litter layer) (*Diószegi et al., 2023*; *Liu, Zhang & Zhang, 2018*). On the other hand, it can also trap, absorb and store water. Some studies have shown that when the rainfall intensity is low, the forest canopy can trap 15% to 40% of the precipitation (*Yali, Xinyu & Zijie, 2024*; *Zhang et al., 2022b*; *Zheng & Jia, 2020*). Therefore, the forest can purify the water quality through these capabilities and prevent the water source from receiving pollution (such as heavy metals, eutrophication) (*Beaulieu, DelSontro & Downing, 2019*; *Li et al., 2022*; *Wang et al., 2024*). However, the increasingly common fragmentation of forest landscapes driven by urbanization, agricultural expansion and industrial development has severely challenged the positive role of forests in water sources, resulting in deterioration of water quality and ecological consequences detrimental to the healthy development of aquatic ecosystems.

Forest fragmentation is the process of dividing large contiguous forest areas into smaller, isolated patches (*Fischer et al., 2021*; *Liu et al., 2019*; *Ma et al., 2023*). This phenomenon not only changes the physical landscape, but also disrupts the ecological processes that sustain terrestrial and aquatic systems (*Fischer et al., 2021*). The impact of fragmentation is not limited to the loss of forest cover, it affects microclimatic conditions, soil properties and hydrodynamics, and ultimately the water quality of rivers (*Clément et al., 2017*). One of the main ways that forest fragmentation affects river water quality is by altering hydrological processes (*Qiu et al., 2023*). Forested areas play a vital role in regulating water flow, intercepting precipitation and facilitating soil infiltration. This natural filtration process reduces surface runoff and allows the water to be gradually released into the river, thereby maintaining flow and water temperature (*Liu, Zhang & Zhang, 2018*; *Vose, 2019*). In contrast, fragmented landscapes often exhibit increased impervious surfaces (such as roads and urban development projects) and bare land, which accelerate runoff and can lead to flash floods (*Sohn et al., 2020*). This rapid runoff transports sediment, nutrients and pollutants directly into the river, seriously affecting water quality. Nutrient load is another major problem associated with forest fragmentation. Forests naturally circulate nutrients to maintain balance and support healthy aquatic ecosystems. However, this balance can be upset when areas of forest are cut down or broken up (*Wenhua, 2004*). For example, increased agricultural activity is often accompanied by deforestation, leading to increased application of fertilizers that may run off into rivers during periods of rainfall (*Brodie & Mitchell, 2005*; *Neill et al., 2013*). Studies have shown that rivers adjacent to fragmented landscapes often experience elevated concentrations of nitrogen and phosphorus, which can lead to eutrophication—a process characterized by excessive nutrient enrichment that fuels algal blooms, depletes oxygen levels, and adversely affects aquatic life (*Chakraborty & Chakraborty, 2021*). At present, it is common to study the impact of landscape characteristics of various land use types on river water quality (*Qiu et al., 2023*; *Wu & Lu, 2021*; *Xiao et al., 2016*), but there are relatively few studies on the impact of forest fragmentation on river water quality.

The upper Ganjiang River basin is a typical mountainous and hilly landform with a forest coverage rate of more than 76%. On the one hand, due to agricultural reclamation and urban expansion, on the other hand, this region has the reputation of "world tungsten capital" and "rare earth kingdom", but it has been affected by extensive mining activities in

the past, resulting in a large area of "gully" and "white desert" landform, among which the surface landscape fragmentation is more serious. The most severely damaged ecosystems are mainly forest lands (*Yan et al., 2016*). Long-term forest fragmentation will affect many aspects including river water quality in this area, so it is necessary and urgent to discuss the impact of forest fragmentation on water quality in the upper Ganjiang River basin. In this context, we took the upper Ganjiang River basin in a typical hilly area as a research area, analyzed the impact of forest landscape fragmentation on river water quality, and determined the threshold of key forest landscape fragmentation indicators leading to abrupt water quality. Our objectives were (1) to assess the health of river water quality; (2) to determine the impact and independent contribution of forest landscape fragmentation metrics on river water quality; (3) to find key thresholds for forest landscape fragmentation metrics that cause sudden changes in water quality.

## MATERIALS AND METHODS

### Study area

The upper Ganjiang River basin is located between latitudes 24°29′N and 27°09′N, and longitudes 113°54′E and 116°38′E (Fig. 1). This region is characterized by a surround formed by the Wuyi, Yu, Zhuguang, Jiulian, and Dayu mountains, resulting in a terrain that exhibits a notable elevation gradient, with higher altitudes predominating in the southern section and lower elevations in the northern part. The landscape is predominantly hilly and mountainous, encompassing approximately 83% of the entire basin. The hydrological system within the basin is characterized by a convergence of water flows, predominantly facilitated by the two major tributaries, the Zhangjiang and Gongjiang Rivers. Upon merging, these tributaries contribute to the formation of the Ganjiang River, which serves as a critical source of both production and domestic water for the inhabitants of southern Jiangxi, particularly in Ganzhou City, Zhanggong District. The basin is situated within a subtropical monsoon climate zone, exhibiting high temperatures and significant rainfall during the summer months, contrasted by cooler temperatures and lower precipitation levels in winter. The mean annual temperature is recorded at 19.8 °C, while the average annual precipitation is approximately 1,318.9 mm.

### Water quality data sources

Water quality data for the 15 monitoring sites (Figs. 1A and 1B) located in the upstream region of the Ganjiang River were sourced from the National Surface Water Quality Dissemination System (www.cnemc.cn/en/) for the periods of January 2021 and July 2021. The study sites are located at the mouths of rivers and are representative of the state of water quality in the sub-basins to which they belong. To ensure seasonal representativeness, data collection involved the selection of four distinct days in both January and July, with one day chosen per week, followed by averaging the values for each water quality indicator (monitoring station water quality data displayed in Table S3). The indicators assessed in this study encompass total phosphorus (TP, mg·L$^{-1}$), total nitrogen (TN, mg·L$^{-1}$), permanganate index (PI, mg·L$^{-1}$), chemical oxygen demand (COD, mg·L$^{-1}$), five-day biochemical oxygen demand (BOD$_5$, mg·L$^{-1}$), and ammoniacal nitrogen (NH$_3$-N,

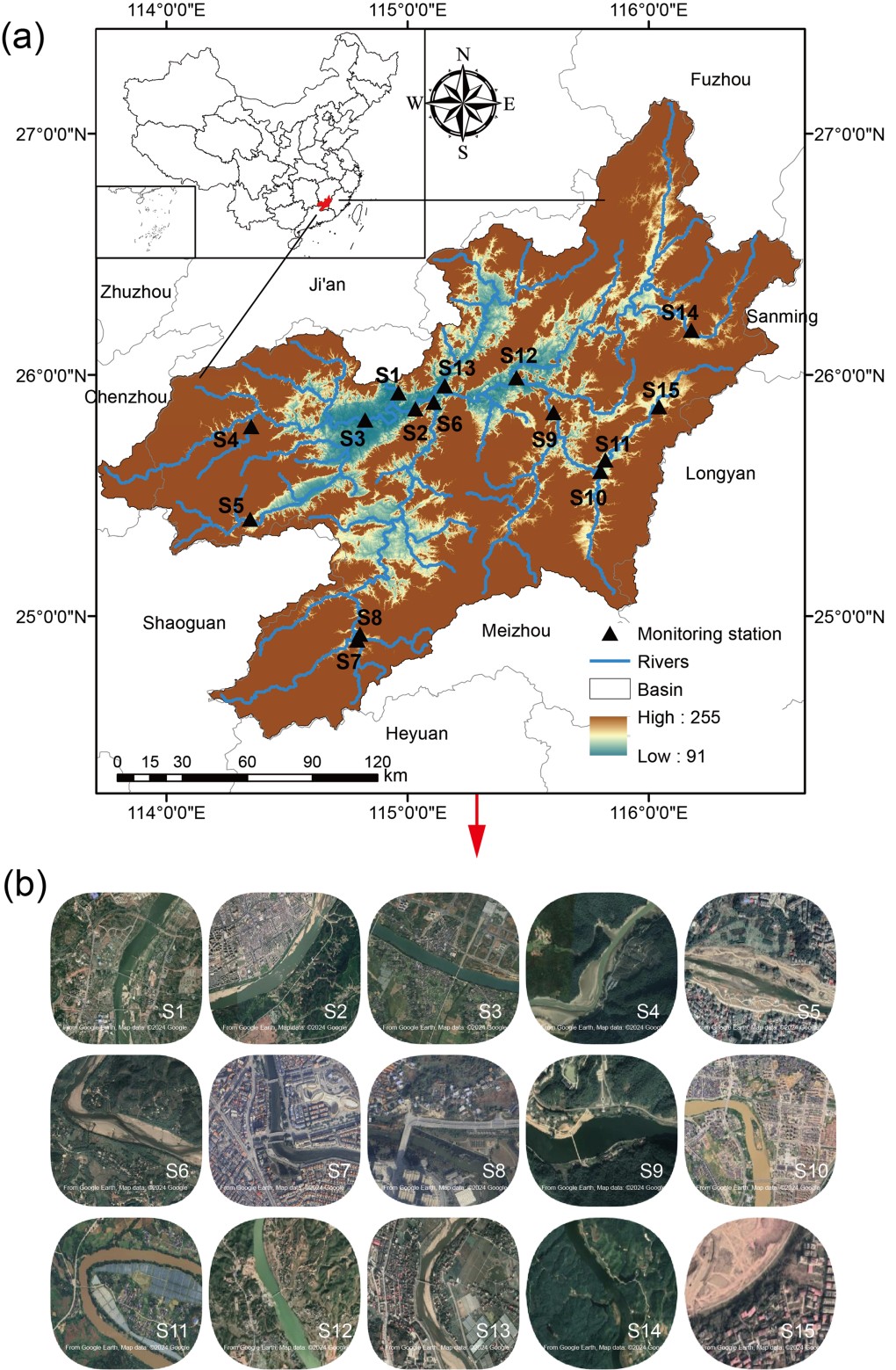

**Figure 1 Study area.** (A) is the distribution of monitoring sites; (B) is the land use around the sites. DEM data from geospatial data cloud (https://www.gscloud.cn); China administrative boundaries data from Database of Global Administrative Areas (https://gadm.org/download_country.html).

mg·L$^{-1}$). For comprehensive information regarding the detection methodologies employed, please refer to the document titled "Water and Wastewater Monitoring and Analysis Methods" (*Committee, 2002*). In addition, the water quality evaluation methodology used in this study is described in the Appendix.

## Forest fragmentation data sources

Data pertaining to the basin and river systems were derived from the Hydrosheds products (https://www.hydrosheds.org/products), the 15 sub-basins were as shown in Fig. 2. The analysis of forest fragmentation utilized land use data with a resolution of 30 m for the year 2021 (https://zenodo.org/records/12779975). All sub-basins are dominated by forest land and cropland, and their forest cover has reached 69.34% to 92.58% (Table S4). Six landscape metrics was selected, including patch density (PD) and mean patch area (MPA), which represent the degree of landscape fragmentation and patch size, respectively; edge density (ED) and interspersion juxtaposition index (IJI), which reflect patch boundary characteristics and the relationships of spatial configuration; and the landscape division index (DIVISION) and effective mesh size (MESH), which measure overall landscape connectivity and fragmentation. These metrics allow for a more comprehensive understanding of the degree of forest landscape fragmentation (*Lin et al., 2024*; *Ma et al., 2023*; *Pfeifer et al., 2017*), which are described in detail as Table 1.

## Statistical analysis

The Canadian Council of Ministers of the Environment Water Quality Index (CCMEWQI) was employed to assess the water quality in the upper reaches of the Ganjiang River, in accordance with the Class III surface water quality standards of China (where the safety thresholds are defined as PI < 6, COD < 20, BOD$_5$ < 4, NH$_3$-N < 1, TP < 0.2, and TN < 1, the exceeding these thresholds indicates that the surface water is unsuitable for drinking). Detailed calculation and evaluation methodologies are provided in Part 1 of the Appendix. To ensure the rigor of our analysis, the normal distribution on all datasets was tested and transformed the data that did not conform to the normal distribution. Furthermore, t-tests were employed to examine significant differences in water quality indicators across varying seasons. Redundancy analysis (RDA) as a methodological tool for elucidating the complex relationship between landscape metrics and water quality highlights its established efficacy within this domain of research (*Clément et al., 2017*; *Wu et al., 2024*; *Xiao et al., 2016*). Prior to the RDA analysis, the water quality indicators underwent Detrended Correspondence Analysis (DCA), revealing that the lengths of all four axes were less than 3, thereby validating the applicability of RDA to this dataset (*Wu & Lu, 2021*). RDA was analyzed with CANOCO 5.0 software (Microcomputer Dynamics, Inc., Metairie, LA, USA). The nonparametric change-point analysis (nCPA) was used to explore thresholds for landscape forest fragmentation metrics that lead to abrupt water quality changes. Statistically significant ($p < 0.05$) change points identified by the nCPA were determined by a permutation test (Bootstrap test method) and 95% confidence intervals for the change points were estimated to indicate that abrupt changes in nitrogen concentrations were associated with landscape fragmentation

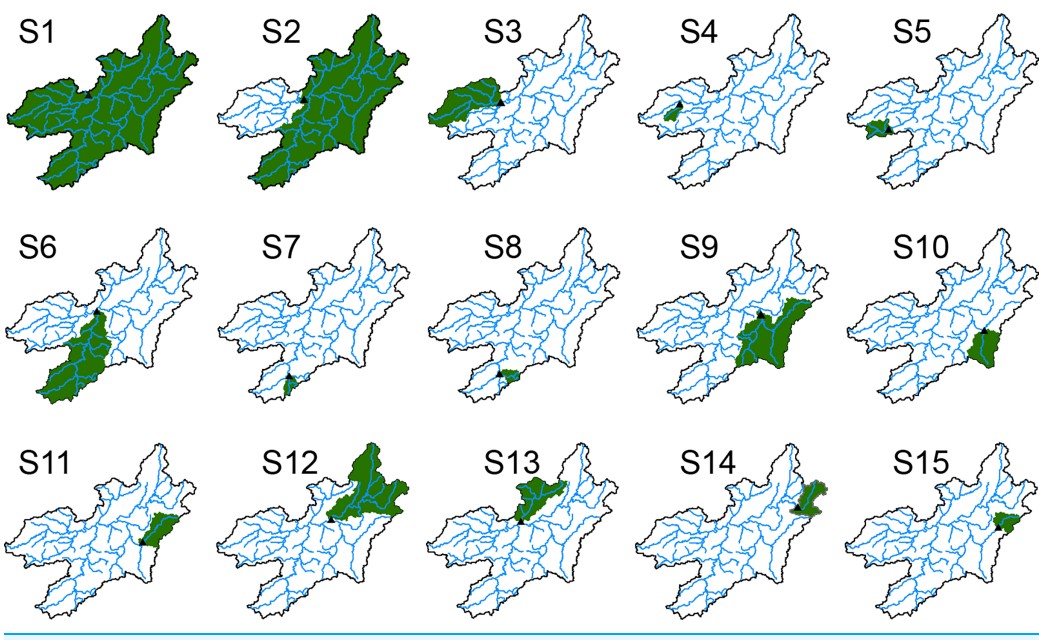

**Figure 2 Monitoring sites and the sub-basins which they represent.**

thresholds rather than random fluctuations (detailed calculations were in Part 2 of the Appendix). The above statistical analysis methods are all completed using R language, except RDA.

# RESULT

## Information and evaluation of water quality in the upper Ganjiang River basin

The basic information, spatial distribution characteristics and water quality evaluation of the six water quality indicators in the upper Ganjiang River basin were shown in Table 2, Figs. 3 and 4, respectively. It was found that there were significant differences in BOD$_5$ and TN concentrations between summer and winter, while no differences in PI, COD, NH$_3$-N and TP (Table 2). The spatial distribution of the water quality indicator showed that the concentrations of PI, COD, BOD and TP at all stations were below the limits of China's Class III surface water quality standards in both winter and summer (green and blue points, see Appendix for detailed description). Concentrations of NH$_3$-N at the S2 monitoring site exceeded the limits of China's Class III surface water quality standards during the winter. For TN, the concentrations at S1 and S11 in summer also exceeded the limits of China's Class III surface water quality standards, and the exceedance was more serious in winter, with some concentrations exceeded the limits of China's Class III surface water quality standards at some sites (S1, S2, S6, S9, and S11) (Fig. 3).

The evaluation of the Canadian Water Quality Index (CCWEWQI) found that the water quality at most of the monitoring sites in the region was generally at good and

**Table 1 Basic description of the forest landscape metrics.**

| Fragmentation metrics | Abbreviation | Unit | Note | Formular | Definition |
|---|---|---|---|---|---|
| Landscape division index | DIVISION | | Responds to the degree of separation or disconnection between landscape patches and is primarily used to assess the degree of landscape fragmentation. | $DIVISION = 1 - \sum_{i=1}^{n} \left(\frac{a_i}{A}\right)^2$ | Where $a_i$ is area of the $i$-th patch of the landscape; $A$ is total landscape area; $n$ is total number of patches in the landscape. |
| Landscape patch density | PD | N/100 ha | Number of patches (e.g., land use types or ecological units) per unit area. | $PD = \frac{N}{A}$ | Where $N$ is total number of patches in the landscape; $A$ is total landscape area. |
| Effective mesh size | MESH | ha | The connectivity between "gaps" in the landscape. | $MESH = \frac{\sum_{i=1}^{n} a_i^2}{A}$ | Where $a_i$ is area of the $i$-th patch of the landscape; $A$ is total landscape area; $n$ is total number of patches in the landscape. |
| Interspersion and Juxtaposition index | IJI | % | The degree of contact (adjacency) and interlacing between patches. | $IJI = \frac{\sum_{i=1}^{m} \sum_{k=i+1}^{m} \left[\left(\frac{e_{ik}}{E}\right) \ln\left(\frac{e_{ik}}{E}\right)\right]}{\ln(0.5 \times m \times (m-1))}$ | Where $e_{ik}$ is length of the edge between patch type $i$ and patch type $k$; $E$ is total length of all edges in the landscape; $m$ is total number of patch types in the landscape. |
| Edge density | ED | m/ha | The relationship between the length of the boundaries of all patches in the landscape and the total area of the landscape. | $ED = \frac{E}{A}$ | Where $E$ total length of edge in the landscape; $A$ is total landscape area. |
| Mean patch area | MPA | ha | The average size of all patches in the landscape, reflecting the degree of landscape fragmentation and patch size distribution. | $MPA = \frac{\sum_{i=1}^{n} a_i}{n}$ | Where $a_i$ is area of the $i$-th patch in the landscape; $n$ is total number of patches in the landscape. |

**Table 2 Descriptive statistics of water quality indicators in the upper Ganjiang River basin.**

| WQI | Summer | | Winter | | Seasonal differences |
|---|---|---|---|---|---|
| | Mean | SD | Mean | SD | |
| PI | 2.11 | 0.53 | 1.85 | 0.89 | |
| COD | 8.27 | 3.22 | 8.18 | 4.48 | |
| BOD$_5$ | 0.77 | 0.41 | 1.34 | 0.54 | ** |
| NH$_3$-N | 0.10 | 0.06 | 0.28 | 0.37 | |
| TP | 0.05 | 0.03 | 0.05 | 0.03 | |
| TN | 1.15 | 0.43 | 1.97 | 0.96 | * |

Notes:
SD, standard deviation.
* Indicates significance at the 0.05 level.
** Indicates significance at the 0.01 level.

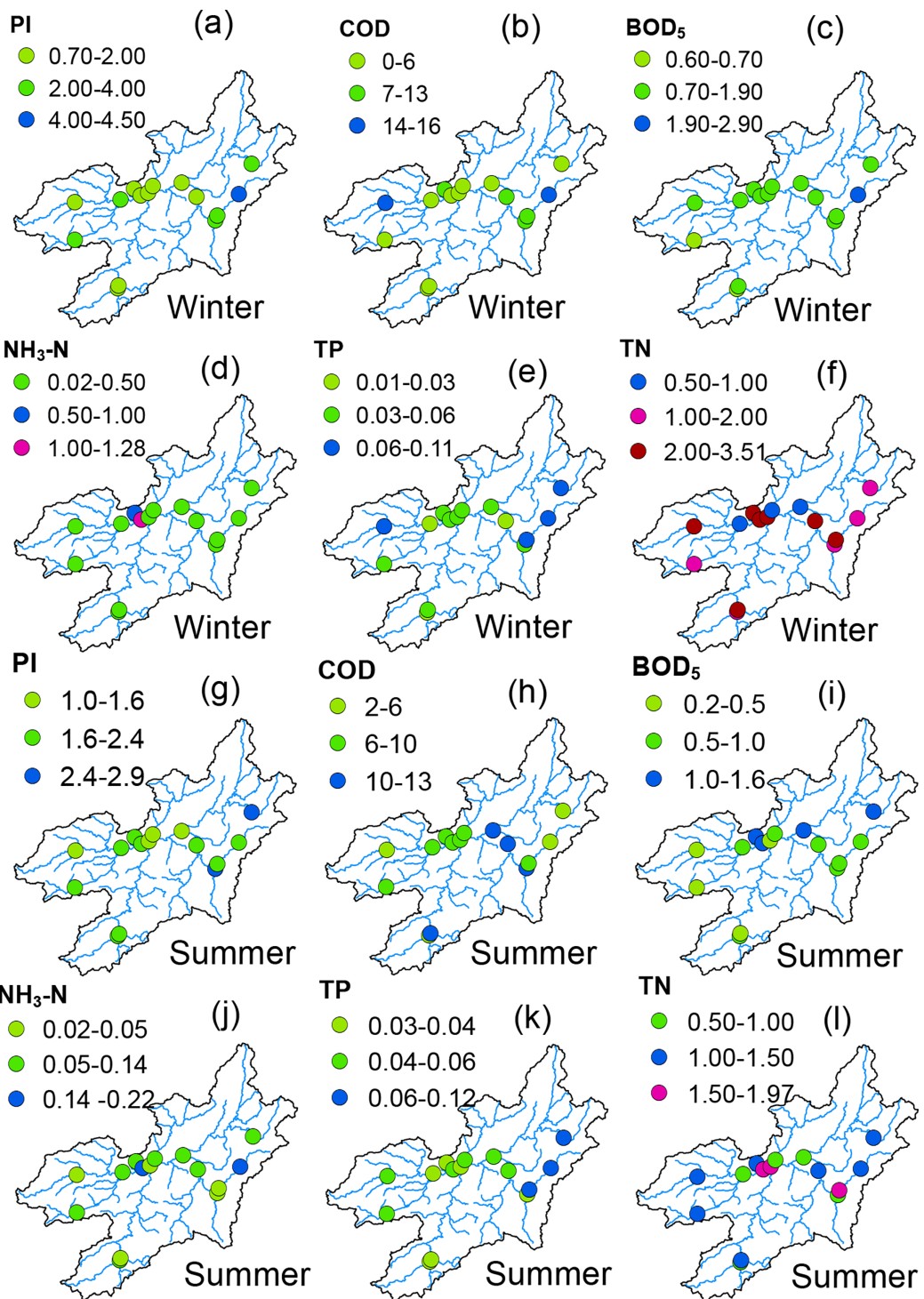

**Figure 3 Spatial distribution of water quality indicators in the upper Ganjiang River basin.** The blue and green dots represent the water quality parameter concentration is lower than the national water quality standard of Class III, while the red dots represent the water quality parameter concentration exceeds the water quality standard of Class III, and the brown dots represent the water quality parameter concentration exceeds the water quality standard of Class V, which is inferior to Class V water quality.

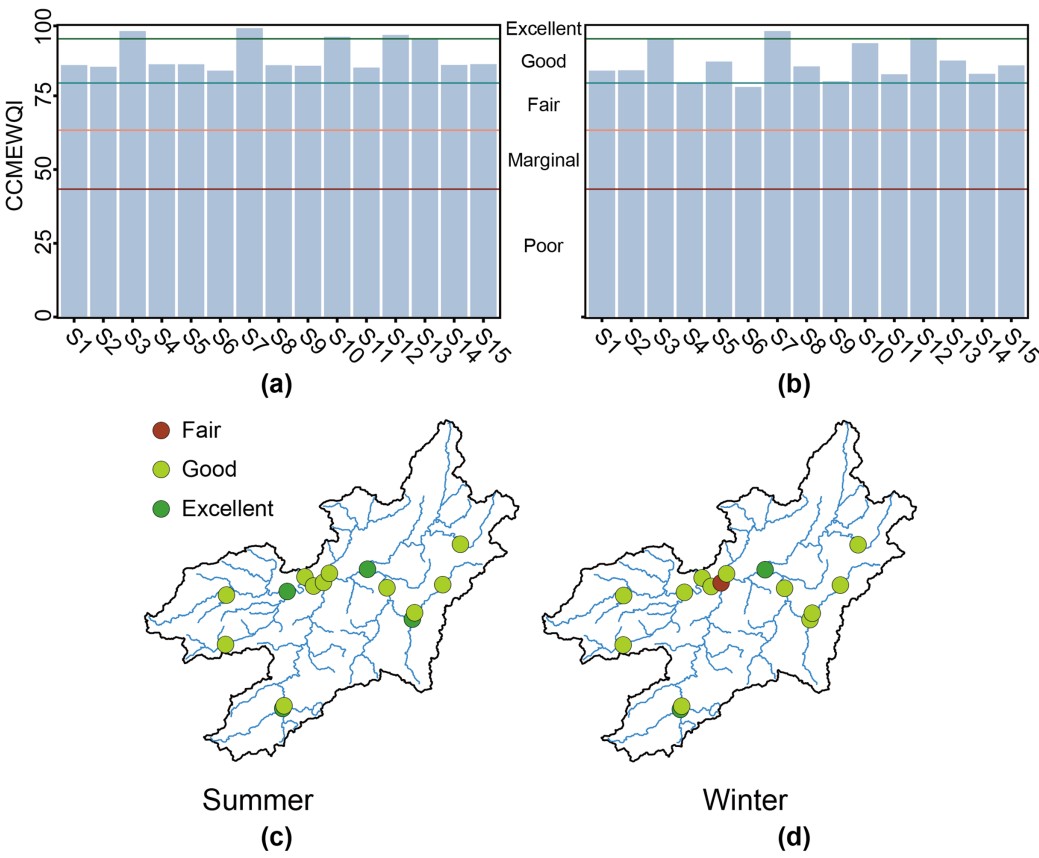

**Figure 4 Evaluation levels (A and B) of water quality at monitoring sites and their spatial distribution (C and D).**

excellent levels (Figs. 4A and 4B), while the water quality at the S4 sample site (located in the urban area of Ganzhou City) was at fair level only in the winter (Fig. 4D).

## Forest landscape fragmentation metrics

The basic information of forest landscape fragmentation metrics in the sub-basins represented by 15 monitoring sites in the upper Ganjiang River basin was presented in Table 3 and Fig. 5. The results indicated that the mean values of PD, ED, MPA, IJI, DIVISION, and MESH for the forested landscapes in the 15 sub-basins were 1.27/100 ha, 31.67 m/ha, 85.17 ha, 21.90%, 0.46, and 241,077.01 ha. The maximum value of PD of the forest occurred at S15 (2.27/100 ha) and minimum value at S4 (0.40/100 ha). The maximum ED value was found at S13 (49.08 m/ha) and minimum value at S4 (15.51 m/ha). The maximum and minimum MPA values were found at S4 (233.06 ha) and S15 (30.49 ha), respectively. The minimum value of IJI was found at S8 (7.57%) and the maximum value was found at S11 (28.82%). The minimum value of DIVISION was found at S4 (0.15) and the maximum value was found at S12 (0.82). Most of the monitoring sites had MESH values below the mean, with a median of only 102,012.14 ha, and only five sites had higher MESH values (S1, S2, S3, S5, S9).

**Table 3  Descriptive statistics of forest landscape fragmentation metrics in the upper Ganjiang River basin.**

|  | Mean | SD | Median | Min | Max |
|---|---|---|---|---|---|
| PD | 1.27 | 0.56 | 1.26 | 0.40 | 2.27 |
| ED | 31.67 | 8.71 | 31.15 | 15.51 | 49.08 |
| MPA | 85.17 | 59.08 | 64.65 | 30.49 | 233.06 |
| IJI | 21.90 | 5.63 | 22.50 | 7.57 | 28.82 |
| DIVISION | 0.46 | 0.21 | 0.46 | 0.15 | 0.82 |
| MESH | 241,077.01 | 274,842.20 | 102,012.14 | 24,064.56 | 869,923.11 |

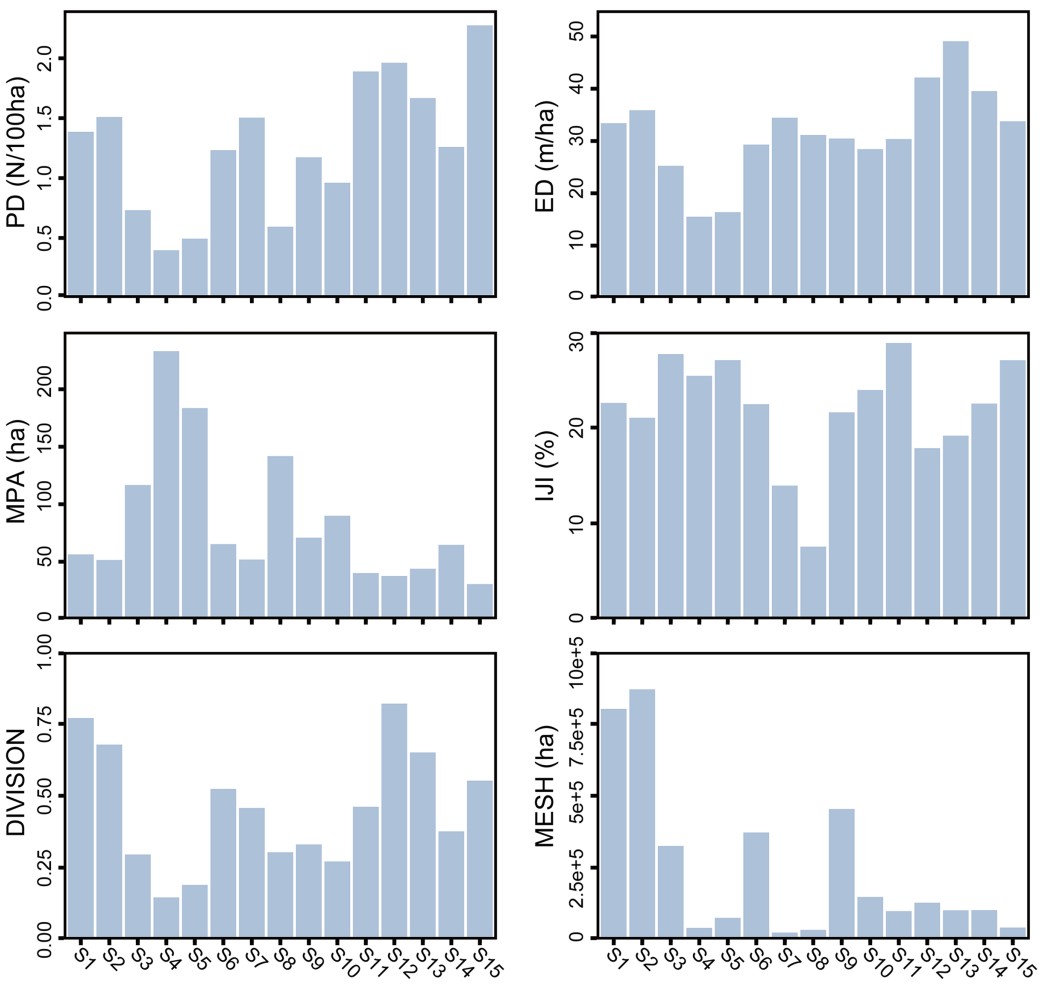

**Figure 5  Forest fragmentation metrics in sub-basins represented by monitoring sites.**

## Relationship between forest landscape fragmentation metrics and water quality

RDA was used to explore the effects of forest fragmentation metrics on overall water quality indicators (Table 4 and Fig. 6). Six forest fragmentation metrics accounted for over 40% of the variance in overall water quality, demonstrating a more pronounced explanatory capacity during the winter season (55.47%) compared to summer (41.21%) (Table 4). This difference highlights the critical role of forest fragmentation in influencing water quality in different seasons.

In summer, MPA and MESH of forest were negatively correlated with most of the river water quality indexes, PD, ED and DIVISION were positively correlated with most of the river water quality indicators, while IJI was only positively correlated with TP, PI and BOD5, and negatively correlated with TN, NH3 and COD. MESH contributed the most (20.9%), followed by DIVISION (20.4%) and IJI the least (6.9%). In winter, IJI, ED, PD and DIVISION were positively correlated with most water quality indicators, while similar to summer, MESH and MPA were negatively correlated with most water quality indicators. IJI had the highest contribution (44.9%) and the lowest contributing fragmentation metrics was MPA (3.9%) (Fig. 7).

## Abrupt change

River water quality was found to be primarily influenced by TN concentrations in previous water quality assessments (Fig. 3). Therefore, nCPA analyzed the thresholds at which forest landscape fragmentation metrics lead to abrupt changes in TN concentrations (Fig. 8). It was found that the values of PD, ED, IJI, DIVISION and MESH were consistent in winter and summer when the probability of abrupt change in TN concentration reached 100%, that is to say they were higher than 1.83/100 ha, 39.4 m/ha, 28.1%, 0.77 and 7.89e +0.5 ha, respectively. However, there was a difference in the thresholds at which MPA led to abrupt changes in TN concentration in winter and summer, with winter (>143.8) the threshold was higher than in summer (>128.9).

# DISCUSSION

## Water quality assessment analysis in the upper Ganjiang River basin

The quality of river water is closely related to the health of the people in the basin and the sustainable development of production and life (*Ibáñez & Peñuelas, 2019*; *Ma et al., 2020*; *Pfeifer et al., 2017*). The CCWEWQI evaluation method was utilized to analyze the overall status of water quality at various monitoring sites (Fig. 4). The results found that the river water quality in the upper Ganjiang River Basin was generally at a good level (most of the monitoring sites had good and excellent water quality), which is similar to the results of previous studies (*Chen et al., 2021*; *Peng, Mingdong & Shaonan, 2023*; *Song et al., 2024*). However, the water quality at monitoring site S6 was assessed as fair in winter, which may be due to the fact that this monitoring site is located in the suburban area of Ganzhou City. On the one hand, there are frequent agricultural activities in the suburban area, and the excessive use of chemical fertilizers and pesticides can lead to the entry of these chemicals into the water body through surface runoff or groundwater infiltration, resulting in

**Table 4 Results of the forest landscape fragmentation metrics for explaining water quality.**

| Season | Statistic | Axis 1 | Axis 2 | Axis 3 | Axis 4 |
|---|---|---|---|---|---|
| Summer | Eigenvalues | 0.23 | 0.09 | 0.07 | 0.03 |
| | Explained variation | 23.15 | 31.65 | 38.23 | 41.21 |
| | Pseudo-canonical correlation | 0.61 | 0.78 | 0.69 | 0.59 |
| Winter | Eigenvalues | 0.37 | 0.11 | 0.05 | 0.02 |
| | Explained variation | 37.36 | 47.9 | 53.2 | 55.47 |
| | Pseudo-canonical correlation | 0.82 | 0.76 | 0.72 | 0.55 |

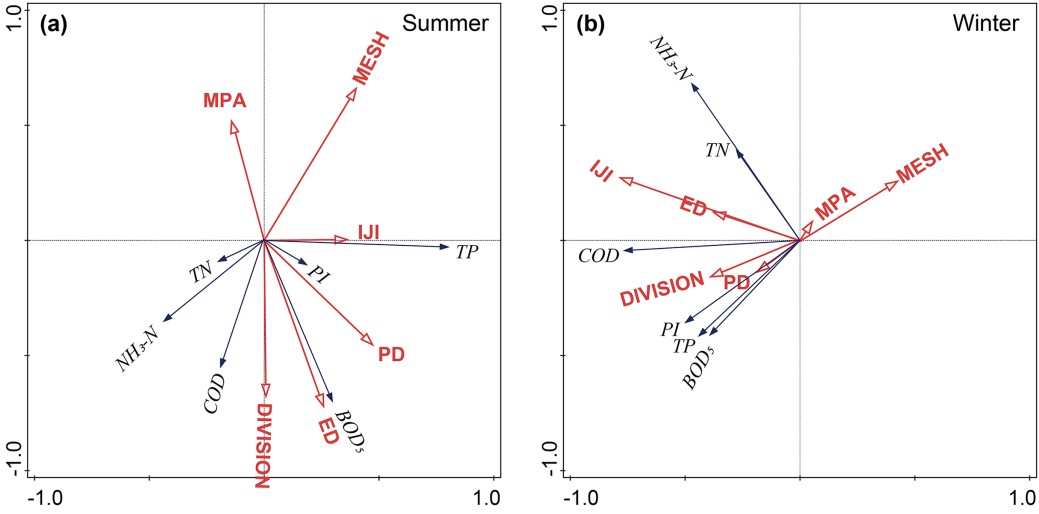

**Figure 6 Relationship between forest landscape fragmentation metrics and water quality indicators in the upper Ganjiang River basin.**

eutrophication and other pollution problems; on the other hand, with the expansion of the city and the increase of population in the suburban area, the discharge of domestic wastewater also increases. If sewage treatment facilities are inadequate or have insufficient capacity, the direct discharge of domestic sewage into water bodies can lead to water quality pollution. For individual water quality indicators, the concentrations of PI, COD and $BOD_5$ in the river of the upper Gangjiang River basin were all within safe limits (not exceeding the Class III water quality standard) (Fig. 3). The concentrations of these three indicators can directly or indirectly represent the concentration of organic matter in the river, and the lower their concentrations, the lower the content of organic matter in the river. Generally speaking, the sources of organic matter in rivers include three main aspects, which are domestic sources (*e.g.*, detergents), agricultural sources (*e.g.*, fertilizers, pesticides, agricultural fertilizers), and industrial sources (*e.g.*, paper, tanneries, petrochemicals) (*Aguilar-Torrejón et al., 2023*; *Singh, Singh & Singh, 2021*). The low river organic matter in this watershed may be due to the low organic matter content of the red soil, which does not allow for the input of excess organic matter into the river, in addition

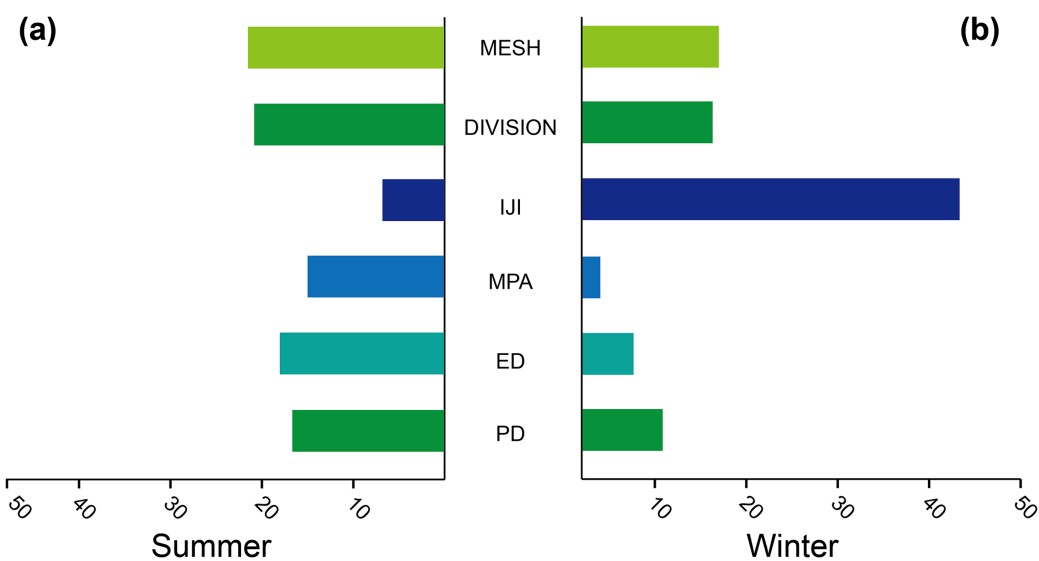

**Figure 7 Contribution of the forest landscape fragmentation metrics.**

to the small area of cultivated land and the low input of organic matter from agricultural sources. Nitrogen and phosphorus are important indicators of eutrophication in water bodies (*Akinnawo, 2023*). The concentrations of TP were low at the monitoring points in this study, again less than the Class III water quality standard. However, high nitrogen concentrations are common for the region, especially in winter, for example, all five monitoring points (S1, S2, S6, S9, and S11) exceeded the Class V water quality standard, of which three monitoring points (S1, S2, and S6) were mainly located within the urban area of Ganzhou City, and thus focused on the TN (production and domestic sources) inflow around the monitoring points, in addition to receiving the TN inflow from the upper Ganjiang River basin. This suggested that TN was a key factor affecting river water quality in this study area, and for this reason, we specifically focused on TN as a research object to analyze the thresholds for abrupt changes in TN concentrations due to forest landscape fragmentation.

## Impacts of forest landscape fragmentation metrics on water quality

Generally, forests play a highly positive role in purifying river water quality. Both trees and shrubs can significantly contribute to the interception of runoff pollutants, and they can also absorb contaminants, including nutrients, through their root systems (*Ellison et al., 2017*; *Qiu et al., 2023*). However, with the increasing intensity of human activities, forests are gradually becoming fragmented (*Fischer et al., 2021*; *Liu et al., 2019*). Research indicated that the fragmentation of forests in most temperate and subtropical regions—primarily in northern Eurasia and South China—has declined between 2000 and 2020 (*Ma et al., 2023*). The fragmentation of forest landscapes (*e.g.*, ED, PD) has been widely recognized as a contributing factor to the deterioration of river water quality (*Clément et al., 2017*; *Qiu et al., 2023*). Therefore, this study delves into the impact of forest landscape

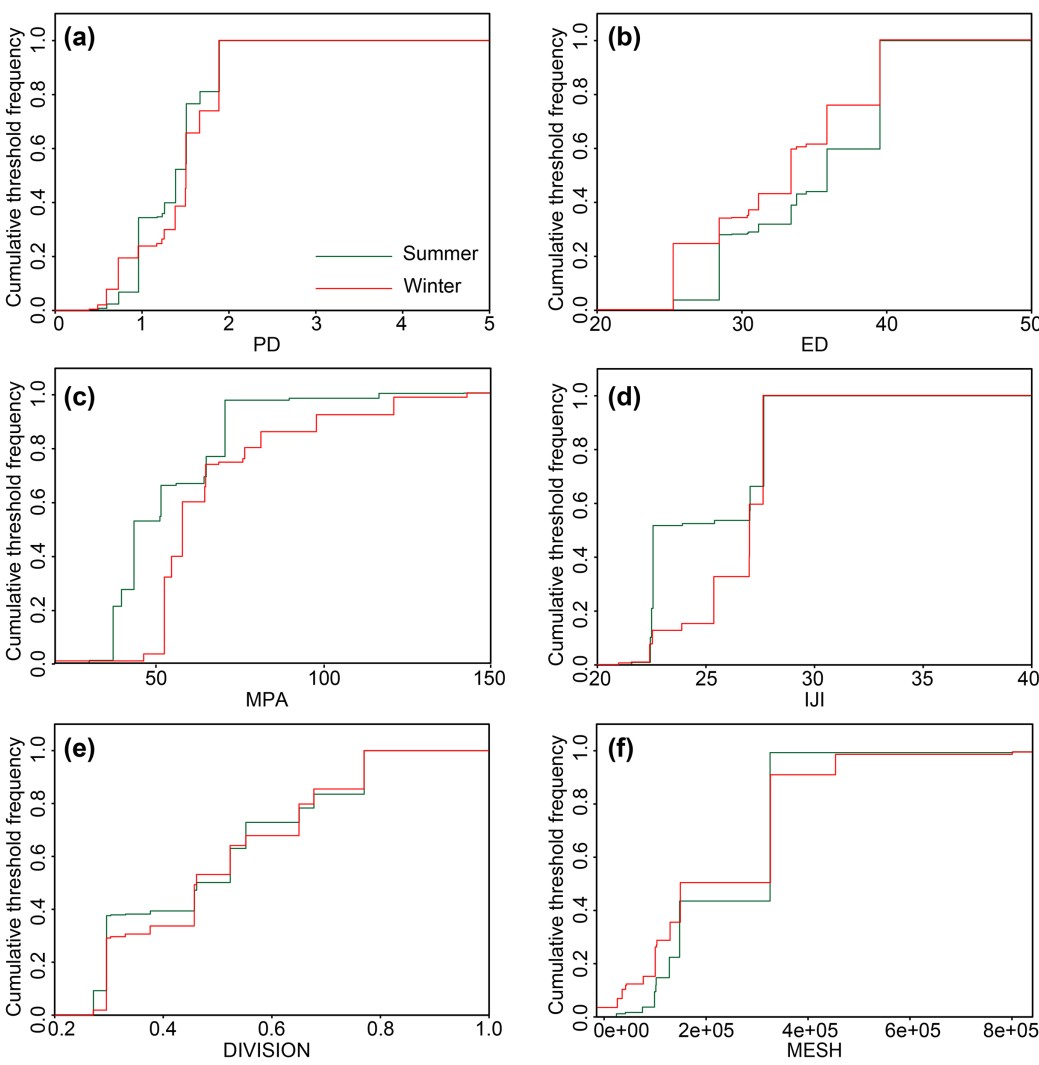

**Figure 8 The cumulative distribution of the changepoint positions using nCPA.** Cumulative threshold frequency for (A) PD, (B) ED, (C) MPA, (D) IJI, (E) DIVISION, (F) MESH. The red line represents winter and the green line represents summer.

fragmentation on the water quality of the upper Ganjiang River basin, which is characterized by high forest cover (>76%), during both winter and summer seasons.

Research indicated that six forest landscape fragmentation metrics can explain 41.21% of the variability in river water quality during the summer; however, the explanatory power increases by 14.26% for winter water quality changes (Table 4). This difference is primarily attributed to local climate and topography. The upper Ganjiang River basin is characterized by a subtropical monsoon climate, with high temperatures, abundant rainfall, and frequent storms during the summer (*He & Liu, 2016*). In a fragmented forest landscape, the interception capacity of the forest is significantly reduced, and the steep hilly terrain may exacerbate the entry of surface pollutants into the river. Conversely, during the winter, precipitation is lower, and the surface runoff's erosion capacity is far less than in summer (*Liu, Zhang & Zhang, 2018*; *Sun et al., 2024*).

In this study, MESH was negatively correlated with most river water quality indicators, and its contribution is 20.9% in summer and 16.6% in winter (Figs. 6 and 7). The MESH reflected the size of the area of unblocked patch types in the landscape. The larger the MESH value, the more connected the patch, the lower the degree of fragmentation, and the better the ecological connectivity (Fischer et al., 2021; Lin et al., 2024). It is shown that the increase of forest MESH value is beneficial to river water quality in the upper reaches of Ganjiang River basin, which is rarely found in related studies. The difference in the MESH value between winter and summer may be due to the size of the canopy. In summer, the forest is rich in foliage, and the canopy of trees is higher than that of winter (Primicia et al., 2013). IJI was only positively correlated with TP, PI and $BOD_5$ in summer water quality indicators (the contribution rate was 6.9%), while it was positively correlated with most river water quality indicators in winter (the contribution rate was as high as 44.9%) (Figs. 6 and 7), indicating that IJI had a higher impact on winter water quality. Generally speaking, the higher the IJI value of the forest, the higher the degree of interleaving between the land use types in space, and the forest is disturbed by other land use, and the boundary is more complex (Wu & Lu, 2021). Different from this study, Wu & Lu (2021) showed that IJI of riparian buffer forest was negatively correlated with water quality indicators in dry season, which may be related to the difference of water quality parameters and research scale. In the two seasons, PD, ED and DIVISION were positively correlated with most of the water quality indicators, indicating that the higher their value, the greater the adverse impact on river water quality (Figs. 6 and 7). This result was also reflected in previous studies (Clément et al., 2017; Shehab et al., 2021; Zhang et al., 2022a). In general, forest fragmentation has a profound impact on the water quality of rivers in the region, and it is extremely important to strengthen the management of forests to avoid excessive disturbance.

## Abrupt change analysis and management recommendations

Abrupt change analysis was conducted to investigate the key thresholds of six forest landscape fragmentation metrics that led to abrupt changes in TN concentrations of the upper Ganjiang River basin. It was found that the thresholds of five forest landscape fragmentation metrics, namely PD, ED, IJI, DIVISION and MESH, leading to abrupt changes in TN concentration in both seasons (winter and summer) were consistent, which indicated that there was no obvious seasonality in the thresholds of these variables affecting TN abrupt changes (Fig. 8). In this study, MESH and DIVISION of forests were the top contributors to the summer water quality change with 20.9% and 20.4%, respectively, but their impacts on water quality were completely opposite, and an increase in the value of MESH favored an increase in the river water quality, while an increase in the value of DIVISION led to a deterioration in the river water quality. Therefore, based on the thresholds for the mutation analysis, it is recommended that in the Upper Ganga River basin forest should exceed 7.89e+0.5 ha for MESH and on the contrary, the value of DIVISION should be controlled within 0.77. In winter, the IJI contribution of forests amounted to 44.9%, which had a much greater impact than the other five forest fragmentation metrics, and since it was positively correlated with most of the water quality

indicators, it is necessary to recommend that the IJI value of forests in the upper Ganjiang River basin must be less than 28.1%. Given that the thresholds for MESH, DIVISION and IJI have no seasonal effect, the thresholds are applicable in both winter and summer. Although nonparametric change point analysis (nCPA) identifies fixed thresholds for landscape fragmentation metrics, it was recognized that these thresholds may oversimplify the reality of changes in water quality. Changes in water quality are often influenced by a combination of factors, including climatic conditions, land use types, and human activities. Therefore, fixed thresholds should be viewed as an initial reference rather than an absolute standard of judgment.

## CONCLUSIONS

The effects of forest landscape fragmentation metrics on river water quality were pronounced and seasonal. The cumulative explanatory ability of the forest landscape fragmentation metrics on water quality was better in winter than in summer. In the upper Ganjiang River basin, MESH and IJI are the most important forest fragmentation metrics affecting water quality in summer and winter, respectively. River water quality will be further improved when MESH >7.89e+0.5 ha in summer and IJI was <28.1% in winter. it is necessary to analyze the effects of forest landscape fragmentation on river water quality to identify its key influences. We concluded that when management or planning with the goal of improving river water quality is undertaken, it can be more effective in protecting river water quality by adequately incorporating abrupt change points in key forest landscape fragmentation metrics in the region.

## ACKNOWLEDGEMENTS

The authors would like to thank anonymous reviewers for their valuable contributions.

### Funding

This work was supported by Scientific Research Program for High-level Talents of Gannan Medical University (No. QD202422), the Science and Technology Research Project of the Jiangxi Provincial Department of Education (No. GJJ2401307), the Talent Development Program for Young Scientific and Technological Professionals in the Early Career Stage in Jiangxi Province, and Research and Innovation Project for Postgraduate in Hunan Province (No. CX20220720). The funders had no role in study design, data collection and analysis, decision to publish, or preparation of the manuscript.

### Grant Disclosures

The following grant information was disclosed by the authors:
Scientific Research Program for High-level Talents of Gannan Medical University: QD202422.
Science and Technology Research Project of the Jiangxi Provincial Department of Education: GJJ2401307.

Talent Development Program for Young Scientific and Technological Professionals in the Early Career Stage in Jiangxi Province.

Research and Innovation Project for Postgraduate in Hunan Province: CX20220720.

## Competing Interests

The authors declare that they have no competing interests.

## Author Contributions

- Biao Li conceived and designed the experiments, analyzed the data, prepared figures and/or tables, and approved the final draft.
- Xiaolei Huang conceived and designed the experiments, performed the experiments, analyzed the data, authored or reviewed drafts of the article, and approved the final draft.
- Qiang Zhong conceived and designed the experiments, performed the experiments, analyzed the data, authored or reviewed drafts of the article, and approved the final draft.
- Xiuxiu Wu analyzed the data, prepared figures and/or tables, authored or reviewed drafts of the article, project support, and approved the final draft.

## Data Availability

The raw measurements are available in the Supplemental File.

## Supplemental Information

Supplemental information for this article can be found online at http://dx.doi.org/10.7717/peerj.19435#supplemental-information.

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
