# Peer review of "Impact of forest fragmentation on river water quality: an example from a typical subtropical hilly basin"

_PeerJ, doi:10.7717/peerj.19435_

## Round 0.1 · original submission · Minor Revisions

There are some helpful comments from a pair of reviewers I'd like you to address. Most of them are fairly minor, but I think they would improve the framing of the overall manuscript.

Reviewer 1 ·

Basic reporting

Criteria 1: Clear and unambiguous, professional English used throughout.
Yes, the language is clear for the most part, apart from a few minor grammatical errors here or there.

Criteria 2: Literature references, sufficient field background/context provided.
Yes, literature citation and referencing is good and provides sufficient context.

Criteria 3: Professional article structure, figures, tables. Raw data shared.
The article structure is fairly standard, and web links to the data sources have been provided. My only concern was with the structuring of the abstract. I see that it has been divided into subsections. However, the other papers I checked in this journal (a small sample size) had a traditional single paragraph abstract. If the journal allows for subsectioned abstracts, then there is no issue.

Criteria 4: Self-contained with relevant results to hypotheses.
Yes, the results are well connected to the hypotheses.

Experimental design

Criteria 1: Original primary research within Aims and Scope of the journal.
Yes, I think this paper falls within the journal's aims and scope.

Criteria 2: Research question well defined, relevant & meaningful. It is stated how research fills an identified knowledge gap.
Yes

Criteria 3: Rigorous investigation performed to a high technical & ethical standard.
Yes, but see my suggestions in the additional comments.

Criteria 4: Methods described with sufficient detail & information to replicate.
Yes, but see my suggestions in the additional comments.

Validity of the findings

Criteria 1: Impact and novelty not assessed. Meaningful replication encouraged where rationale & benefit to literature is clearly stated.
I won't comment on the impact, but the research question asked is valid, and the work conducted seems fairly novel and robust.

Criteria 2: All underlying data have been provided; they are robust, statistically sound, & controlled.
Mostly; see my suggestions in the additional comments.

Criteria 3: Conclusions are well stated, linked to original research question & limited to supporting results.
Yes, the conclusions are well-linked to the research question and the presented results.

Additional comments

Overall, the study presented in this manuscript is well-motivated and contains good analysis of the water quality data and forest fragmentation metrics. I have the following suggestions to the authors to improve its presentation:
1) In the methods section, the land use distribution within the contributing catchments of each sampling site needs to be provided. Currently, there is a brief mention in the Introduction that 76% of the catchment is forested, which I don't think is sufficient.
2) Again in the methods section, more details are needed about the forest fragmentation metrics, especially their mathematical formula and how they were calculated. The information in Table 1, although useful, is not sufficient. It would also be good to provide a rationale for choosing these six landscape metrics.
3) Because the land use distribution information has not been provided, it was not possible to understand why the water quality variations were occurring at different sampling sites. One that information has been added, I think the authors should analyse their water quality results within the context of land use distribution. Although forest fragmentation is important, other land uses might also be a major contributing factor to the spatial variations seen in the results.

·

Basic reporting

No Comments

Experimental design

No Comments

Validity of the findings

No Comments

Additional comments

Comments to Authors
The manuscript titled "Impact of forest fragmentation on river water quality: an example from a typical subtropical hilly basin" explores the relationship between forest fragmentation and river water quality using data from the upper Ganjiang River basin. The study utilizes water quality data from 15 monitoring sites collected in winter and summer, combined with forest landscape fragmentation indices, to assess seasonal variations in water quality. The study employs Redundancy Analysis (RDA) to determine the explanatory power of forest fragmentation on water quality and identifies key fragmentation indices that influence nitrogen pollution. The results highlight the role of effective mesh size (MESH) and interspersion and juxtaposition index (IJI) in regulating seasonal water quality, offering insights into the management of forested watersheds to mitigate water quality degradation.
The manuscript presents a well-structured study on the impact of forest fragmentation on river water quality in the upper Ganjiang River basin. Its key strengths lie in the integration of water quality data with forest fragmentation metrics across seasons, providing valuable insights into temporal variations. The use of Redundancy Analysis (RDA) and nonparametric change-point analysis (nCPA) strengthens the statistical framework, effectively identifying critical fragmentation metrics such as effective mesh size (MESH) and interspersion and juxtaposition index (IJI), which influence water quality patterns and offer guidance for watershed management.
However, the study has limitations. Its focus on a single river basin limits broader applicability, and while the statistical analysis is rigorous, it does not establish direct causal links between fragmentation and water quality. Additionally, the role of anthropogenic activities, such as urbanization and agriculture, is not fully explored, which could provide more context to the findings.
Given the study’s strong methodology and relevance, the manuscript is suitable for publication with minor revisions to improve clarity, expand on human influence, and refine the interpretation of statistical findings.
Comments:
1. The abstract effectively summarizes the study, but some sentences could be more concise. For example, the phrase "the index of forest landscape fragmentation in the sub-basin was calculated to explore the impact of forest fragmentation on river water quality in different seasons." could be reworded for better readability.
2. The sentence "Increasing fragmentation of forests caused by natural and man-made drivers." is incomplete. It should be revised to "Forest fragmentation, driven by natural and human activities, is increasing.
3. The phrase "water quality in the area is general satisfactory" should be corrected to "water quality in the area is generally satisfactory.
4. The abstract presents the major findings well, but the explanation of threshold values (e.g., "the thresholds for IJI and MESH of forest were the same in winter and summer, 28.1% and 7.89e+0.5ha") might be unclear to a general audience. Briefly clarifying the implication of these thresholds would improve accessibility.
5. The abstract concludes with a statement on the study’s significance, but it would be beneficial to explicitly mention how these findings contribute to forest and water resource management beyond theoretical implications.
6. The authors should clarify whether this basin is representative of subtropical hilly watersheds and whether the observed forest fragmentation levels are unique or generalizable to similar landscapes.
7. The study states that water quality data were collected from 15 monitoring sites in January (winter) and July (summer). However, it does not provide justification for:
 Why these specific sites were selected.
 Whether the sample size is statistically representative of the entire basin.
 The spatial distribution of the sites (e.g., upstream vs. downstream).
8. Why RDA was chosen over other multivariate techniques (e.g., Principal Component Analysis (PCA), Structural Equation Modeling (SEM)).
9. Whether the assumptions for RDA (e.g., linear relationships, normal distribution) were checked. If checked should be mentioned either in supplementary portion for better understanding of the underling phenomena.
10. The manuscript uses nonparametric change-point analysis (nCPA) to identify fragmentation thresholds leading to abrupt changes in nitrogen concentration. However:
 The manuscript does not explain whether the identified thresholds are statistically significant or if confidence intervals were estimated.
 The results suggest fixed thresholds (e.g., MESH > 7.89e+0.5 ha, IJI > 28.1%), but this may oversimplify real-world variations in water quality.
 It does not consider potential confounding factors such as precipitation patterns, land-use history, or soil properties.
11. The study does not mention any sensitivity analysis to assess the robustness of results. If different fragmentation indices were used, would the results still hold? Would the findings change if a different statistical method were applied?
12. (Results): The section presents descriptive statistics for six water quality indicators but lacks a clear reference to acceptable environmental thresholds beyond Class III standards. Without proper contextualization, the reader may not fully grasp whether the observed values indicate significant pollution, seasonal variability, or background conditions.
13. The result states that water quality differs significantly between summer and winter, particularly for BOD₅ and TN, but lacks a robust statistical validation beyond basic t-tests. Simple t-tests may not capture complex seasonal trends, particularly if interactions with other factors (e.g., precipitation, land use) are ignored. Use ANOVA or mixed-effects models to confirm whether seasonal differences are statistically significant while controlling spatial variation.
14. The study finds that MESH (effective mesh size) negatively correlates with river pollution while IJI (interspersion and juxtaposition index) is a key winter determinant. However, this interpretation is purely correlation-based. Correlation does not imply causation. There may be confounding variables (e.g., land use type, soil characteristics, precipitation patterns) influencing this relationship.
15. While the study describes mean values of water quality indicators, it does not adequately discuss spatial patterns (e.g., are degraded sites clustered in urban/agricultural areas?). Identifying spatial hotspots of pollution could provide more actionable insights. Use spatial regression models or geostatistical techniques (e.g., kriging) to better analyze spatial variability.
This study offers important insights into how forest fragmentation affects river water quality. Improving the methodology, sharpening the analysis, and clarifying the results will make it even stronger. A clearer discussion on seasonal differences, spatial patterns, and management recommendations will add more depth. Good luck with your revisions!

---

## Round 0.2 · accepted · Accept

Thank you for addressing all of the reviewer comments. I reviewed the manuscript myself and found it ready for publication.